# Small-Signal Stability Analysis and MOSMA-Based Optimization Control Strategy of OWF with MMC-HVDC Grid Connection

**DOI:** 10.3390/s24010139

**Published:** 2023-12-26

**Authors:** Jie Zheng, Hui Li, Bo Zhang, Qinghe Li

**Affiliations:** 1State Key Laboratory of Power Transmission Equipment and System Security and New Technology, School of Electrical Engineering, Chongqing University, Chongqing 400044, China; cqulh@163.com (H.L.); 202211131102t@stu.cqu.edu.cn (B.Z.); Lqh_213@163.com (Q.L.); 2CSSC Haizhuang Windpower Co., Ltd., Chongqing 400044, China

**Keywords:** offshore wind farm, MMC-HVDC, small-signal stability, control strategy, parameter optimization, multi-objective slime mold algorithm

## Abstract

The recent oscillation events in offshore wind farms (OWFs) connected via a modular multilevel-converter-based HVDC (MMC-HVDC) system are developing towards a wider frequency band, which causes complex a small-signal interaction phenomenon and difficulties in the stability analysis and control. In this paper, the wideband dynamic interaction mechanism is investigated based on the impedance analysis method and an improved control strategy using an optimization algorithm is proposed to improve the small-signal stability and reduce the oscillation risks. First, the detailed impedance models of the grid-connected system are established considering the distribution characteristics of the submarine cable, control delay and frequency coupling effect. Then, combined with the active damping control method, the wideband resonance mechanism is analyzed, and the stability constraints of controller parameters are obtained using the impedance stability criterion. Finally, an improved multi-objective slime mold algorithm (MOSMA)-based coordinated optimization control strategy is proposed to enhance the adaptability of the controller parameters and the wideband damping ability of a grid-connected system, which can improve the wideband stability of the system. The simulation and experimental results verify the proposed control strategy.

## 1. Introduction

When AC transmission technology is used in offshore wind farms, the capacitive effect of long-distance AC submarine cables restricts the capacity and distance of line transmission, reactive power compensation devices need to be set up at intervals, which limits the transmission distance, and it is difficult to adapt to the needs of wind energy development in deep sea areas. Recently, due to the resource advantages of offshore wind power and the technical advantages of the modular multilevel-converter-based high-voltage direct current (MMC-HVDC) transmission system [1], a large number of MMC-HVDC-based offshore wind farm (OWF) integration projects are being constructed or operated [2], showing an integration trend of long distance and large capacity. The wind turbine converter and MMC station are power electronic devices with complex control systems and wideband response characteristics, and the submarine cable is an important part of the internal collection network and external transmission network system of the OWF. The complex dynamic behaviors of grid-connected equipment/opponents and the negative damping interaction between the OWF and the MMC easily lead to wideband oscillations in the grid-connected system, including the sub/super-synchronous and medium/high-frequency small-signal stability issues [3,4,5], which seriously threaten the equipment safety and stable operation of the system. MMC-HVDC grid-connected system of offshore wind farms are affected by the complex interaction between wind turbines, wind farm collector lines and MMC, and the MMC-HVDC grid-connected system of offshore wind farms presents coupling characteristics of low damping and multiple instability modes, and the safety and stability of the grid-connected system face more severe challenges. Therefore, the resonance mechanism of wideband interaction in the grid-connected system and the control method for mitigating the oscillations need to be further investigated.

Researchers have studied the small-signal stability of OWFs and MMC-HVDC systems. The primary stability analysis approaches are mainly divided into time domain and frequency domain analysis methods based on eigenvalue calculations and the impedance model, respectively [6,7]. Compared with the eigenvalue analysis, the impedance analysis method has the advantages of a clear physical meaning, accurate description of external characteristics and strong engineering practicality [8]. The AC-side sequence impedance model of MMC was developed in [9] based on the harmonic state-space (HSS) theory considering the multi-frequency coupling characteristics of MMC’s internal dynamics.

The harmonic linearization-based impedance modeling method can establish the impedance model of a wind power converter by reflecting the frequency coupling effect in the sub/super-synchronous ranges [10]. These studies mainly focus on the modeling of the sub/super-synchronous dynamics in a grid-connected system, and the dominant dynamics in the medium/high-frequency band are simplified in the modeling process, such as MMC control delay [11]. When analyzing the medium/high-frequency resonances (MHFRs) issues, the impedance modeling of MMC usually ignores the dynamics of the AC voltage loop and the circulating current suppression controller (CCSC) [12]. Therefore, the current impedance modeling method easily leads to errors in the wideband small-signal stability analysis. In addition, the capacitance effect of the submarine cable can easily inspire the MHFRs in the OWF integration system [13]. Therefore, the characteristics of the submarine cable need to be accurately modeled for small-signal stability analysis. The multiple π-sections cable model was adopted to analyze the high-frequency distribution features of the AC cable in wind power integration system in [14]. Accordingly, the frequency domain analysis method based on impedance model was used to study the MHFRs in the wind farm collection network in [15]. However, numerous series segments easily cause spurious numerical oscillations and increase computational complexity. Therefore, there is a lack of wideband stability analysis models applicable for MMC and an exact analytical model of the capacitance effect of long submarine cable.

Based on the impedance analysis model, the small-signal stability of the grid-connected system is studied by applying impedance stability analysis tools such as the Nyquist stability criterion [16] and Bode diagram [17]. The sub/super-synchronous stability of the MMC-HVDC-connected OWF was analyzed with different bandwidths of MMC current controller in [17]. The influence of controller dynamics on the wideband (1–1000 Hz) stability of the system was explored in [18]. In the above research, the influence factors in MHFRs are not explained. In Ref. [19], the stability mechanism for the MHFRs between the MMC and AC grid system was examined by considering the control delay effects based on the high-frequency simplified impedance model of MMC, but the influence of the capacitive characteristics of the AC transmission line on harmonic resonances was not thoroughly studied. The influence factors on the MHFRs of the MMC-HVDC system was further investigated in [20], including the current controller, voltage feed-forward loop, and capacitive characteristics of the AC line. However, the instability phenomenon investigated focuses on a specific frequency range and mainly on the MMC-HVDC system. The wideband interaction mechanism and stability problems of the MMC-HVDC-connected OWF are not fully explored.

Impedance reshaping methods have been applied to solve wideband instability issues, including damping control [21,22] and parameter optimization design [23,24]. In Ref. [21], a novel fuzzy control scheme for damping the sub-synchronous resonance (SSR) according to the wide-area measurement system (WAMS) in power systems including doubly fed induction generator (DFIG)-based wind farms connected to series capacitive compensated transmission networks is presented. In Ref. [22], a coordinated control method that requires adding an active controller and passive damper in the MMC system was studied to reduce the negative damping in the high-frequency range and it was observed that the filter introduces negative damping near its cut-off frequency, but the impact of the damping control method on the sub/super-synchronous dynamic stability is not considered. In Ref. [23], a novel robust control approach for a nonlinear clutchless automated manual transmission (CAMT) in pure electric vehicles was developed, considering the issue of inexactly measured scheduling parameters that ensure the stability of the closed-loop system with hybrid continuous- and discrete-time systems. In Ref. [24], based on the impedance analysis method, an optimal design method for AC voltage-controlling parameters of the MMC was proposed to guarantee the sub-synchronous stability of the MMC-based OWF integration system, but there was a lack of modeling analysis and suppression of the MHFRs. Owing to the complex controller characteristics and numerous control parameters in the MMC-HVDC-connected OWF, it is difficult to use the traditional trial-and-error method to design the control system. Additionally, the controller design problem for the grid-connected system can be specified as a multi-objective optimization problem (MOP). The popular algorithms for solving MOPs include the evolutionary non-dominated sorting genetic algorithm (NSGA-III) [25] and multi-objective slime mold algorithm (MOSMA) [26]. The NSGA-III algorithm was applied to design the AC voltage controller parameter of the MMC and form the stability constraints in [25]. However, the algorithm optimization performance needs to be improved when suppressing the wideband oscillations and the stability dominant factors are not comprehensively studied. The MOSMA algorithm employs the crowding distance operator and non-dominated sorting to obtain Pareto optimal solutions [26], which has been applied to different fields to a certain extent [27,28], but the initial population quality, global search and convergence ability of the algorithm remain to be improved, and the applicability for small-signal stability problems needs to be further explored.

As analyzed above, the existing studies on the small-signal stability and control mainly aim at the specific frequency band, which makes it difficult to eliminate the additional negative damping characteristics and wideband oscillation issues caused by mismatched controller parameters or damping controller. In order to overcome the aforementioned problems, the wideband complete impedance model of the grid-connect system is established, the wideband dynamic interaction mechanism is revealed in detail based on impedance stability theory, and an improved stability control strategy combining active damping control and controller parameter optimization algorithm is proposed. The key contributions of this study are summarized as follows:(1)The distributed parameter model is developed to accurately analyze the submarine cable capacitance effect and the equivalent impedance model of the wind farm is established considering wideband influence factors. The wideband impedance model of MMC is also established considering the multi-harmonic characteristics, control dynamics and delay effect based on HSS theory.(2)The capacitance effect of long submarine cable and the delay effect of MMC will weaken the high-frequency damping of the system, and the control system dynamics will increase the risk of sub/super-synchronous resonance. The influence of the MHFRs’ suppression strategy on the wideband dynamic performance of the system is investigated, and the wideband stability constraints for the optimized controller parameter design are evaluated.(3)Considering the design principle of wideband dominant factors, an improved MOSMA (IMOSMA) algorithm is advanced to assist the controller parameter design for both the wind farm and the MMC system, and improve the wideband damping ability of the grid-connected system with a greater comprehensive optimization performance.

The main content of this paper is organized as follows: Section 2 provides the wideband impedance model of the wind farm and MMC. Section 3 introduces the active damping controller, reveals the wideband stability mechanism, and derives the stability constraints for the wind farm and MMC. In Section 4, an IMOSMA-based optimal control strategy is proposed to design controller parameters and improve the wideband stability of the system. Section 5 and Section 6 provide the verifications and a conclusion, respectively.

## 2. Wideband Impedance Modeling of MMC-HVDC-Connected OWF

An accurate mathematical model is a prerequisite for analyzing wideband oscillations in the OWF with MMC-HVDC grid-connected system. In this section, the detailed impedance model of OWF and MMC station are established by using the harmonic linearization method and HSS method, respectively, which consider detailed wideband influence factors. For OWF, the capacitance effect of submarine cables dominated in medium- and high-frequency bands and the dynamic characteristics of current loop and PLL control dominated in sub/super-synchronous-frequency bands are both considered. For MMC, the control delay effect dominated in medium- and high-frequency bands and the V/F control dynamic characteristics dominated in sub/super-synchronous-frequency bands are both considered.

### 2.1. System Configuration

Figure 1 shows the configuration of the MMC-HVDC-connected OWF, comprising wind turbines, the AC submarine cable collection network, the MMC-HVDC system, and control systems. The 690 V output of wind turbines is stepped up to 35 kV for connection with other wind turbine units. Numerous units are connected in series via submarine cables and integrated into a large-scale OWF, then fed into the wind-farm-side MMC (WSMMC) to achieve centralizing power transmission through the 220 kV transmission submarine cable. An interconnected system composed of the OWF with the AC cable collection and WSMMC transmission is studied, where the wind turbines and GSMMC are equivalent to the grid-side converter (GSC) and DC source, respectively.

### 2.2. MMC Impedance Modeling

Figure 2 shows the main circuit of the MMC station. Each arm in MMC comprises an arm reactor and *N*_s_ half-bridge sub-modules (SMs), where *C*_sm_, *L*_arm_ and *R*_arm_ are the submodule capacitance, arm inductance and resistance, respectively. $v_{\mathrm{cuk}}^{\Sigma }$, $v_{\mathrm{cbk}}^{\Sigma }$, *i*_uk_ and *i*_bk_ are the sum of voltages and currents in the upper and lower arms of the SMs, respectively. *m*_uk_ and *m*_bk_ are the upper and lower modulation indexes. *v*_sk_, *i*_sk_, and *i*_ck_ are the phase voltage and current, circulating current in the AC side. *I*_dc_ and *U*_dc_ are the voltage and current in the DC side. The subscript “u/b” denotes the upper and lower arms, and “k” denotes the phase number.

According to the bridge arm average model, the MMC equivalent output voltage is as follows:(1)$$v_{\mathrm{uk}}=m_{\mathrm{uk}}v_{\mathrm{cuk}}^{\Sigma },\;v_{\mathrm{bk}}=m_{\mathrm{bk}}v_{\mathrm{cbk}}^{\Sigma }$$

Considering the dynamic behavior of the charge and discharge currents of the bridge arm sub-module, the state equation of the sum of the capacitance voltages of the upper and lower bridge arm sub-modules of the MMC is as follows:(2)$$\left\{\begin{array}{l} \frac{dv_{\mathrm{cuk}}^{\Sigma }}{dt}=\frac{m_{\mathrm{uk}}}{C_{\mathrm{arm}}}i_{\mathrm{uk}}=\frac{m_{\mathrm{uk}}}{2C_{\mathrm{arm}}}i_{\mathrm{sk}}+\frac{m_{\mathrm{uk}}}{C_{\mathrm{arm}}}i_{\mathrm{ck}}\\ \frac{dv_{\mathrm{cbk}}^{\Sigma }}{dt}=\frac{m_{\mathrm{bk}}}{C_{\mathrm{arm}}}i_{\mathrm{bk}}=-\frac{m_{\mathrm{bk}}}{2C_{\mathrm{arm}}}i_{\mathrm{sk}}+\frac{m_{\mathrm{bk}}}{C_{\mathrm{arm}}}i_{\mathrm{ck}} \end{array}\right.$$

According to the generation mechanism of MMC multi-harmonic coupling characteristics, the equation of state with bridge arm circulation *i*_ck_, the AC current *i*_sk_ as state variables can be obtained, and the theoretical model of MMC AC and DC side is expressed as follows:(3)$$\left\{\begin{array}{l} L_{\mathrm{arm}}\frac{di_{\mathrm{ck}}}{dt}=U_{m}-R_{\mathrm{arm}}i_{\mathrm{ck}}-\frac{m_{\mathrm{uk}}v_{\mathrm{cuk}}^{\Sigma }}{2}-\frac{m_{\mathrm{bk}}v_{\mathrm{cbk}}^{\Sigma }}{2}\\ \frac{dv_{\mathrm{cuk}}^{\Sigma }}{dt}=\frac{m_{\mathrm{uk}}}{2C_{\mathrm{arm}}}i_{\mathrm{sk}}+\frac{m_{\mathrm{uk}}}{C_{\mathrm{arm}}}i_{\mathrm{ck}}\\ \frac{dv_{\mathrm{cbk}}^{\Sigma }}{dt}=-\frac{m_{\mathrm{bk}}}{2C_{\mathrm{arm}}}i_{\mathrm{sk}}+\frac{m_{\mathrm{bk}}}{C_{\mathrm{arm}}}i_{\mathrm{ck}}\\ L_{\mathrm{arm}}\frac{di_{\mathrm{sk}}}{dt}=-2v_{\mathrm{sk}}-R_{\mathrm{arm}}i_{\mathrm{sk}}-m_{\mathrm{uk}}v_{\mathrm{cuk}}^{\Sigma }+m_{\mathrm{bk}}v_{\mathrm{cbk}}^{\Sigma } \end{array}\right.$$

A linear and periodically time-varying model of the AC/DC side of the MMC can be obtained using the average value model theory and the harmonic linearization principle [9]. The core dynamic equation is as follows:(4)$$\left\{\begin{array}{l} \begin{array}{c} d\Delta i_{c}/dt=(-R_{m}\Delta i_{c}-0.5n_{\mathrm{bw}}\Delta v_{\mathrm{cb}}^{\Sigma }-0.5\Delta n_{b}v_{\mathrm{cbw}}^{\Sigma }-\\ \;0.5n_{\mathrm{uw}}\Delta v_{\mathrm{cu}}^{\Sigma }-0.5\Delta n_{u}v_{\mathrm{cuw}}^{\Sigma })/L_{m} \end{array}\\ \begin{array}{c} d\Delta v_{\mathrm{cu}}^{\Sigma }/dt=(n_{\mathrm{uw}}(\Delta i_{c}+0.5\Delta i_{s})+\\ \;\;\;\;\;\;\;\;\;\;\;\;\;\;\;\;\;\;\;\;\Delta n_{u}(i_{\mathrm{cw}}+0.5i_{\mathrm{sw}}))/C_{\mathrm{arm}} \end{array}\\ \begin{array}{c} d\Delta v_{\mathrm{cb}}^{\Sigma }/dt=(n_{\mathrm{bw}}(\Delta i_{c}-0.5\Delta i_{s})+\\ \;\;\;\;\;\;\;\;\;\;\;\;\;\;\;\;\;\;\;\;\Delta n_{b}(i_{\mathrm{cw}}-0.5i_{\mathrm{sw}}))/C_{\mathrm{arm}} \end{array}\\ \begin{array}{c} d\Delta i_{s}/dt=(-R_{m}\Delta i_{s}-2\Delta v_{s}+n_{\mathrm{bw}}\Delta v_{\mathrm{cb}}^{\Sigma }+\Delta n_{b}v_{\mathrm{cbw}}^{\Sigma }-\\ n_{\mathrm{uw}}\Delta v_{\mathrm{cu}}^{\Sigma }-\Delta n_{u}v_{\mathrm{cuw}}^{\Sigma })/L_{m} \end{array} \end{array}\right.$$
where “Δ” represents the small-signal components of a different electrical quantity and the subscript “w” represents the steady-state components of a different electrical quantity. *C*_arm_ = *C*_sm_/*N*_s_ is the equivalent capacitance of arm.

According to Laplace transform and harmonic balance principle, Equation (4) can be rewritten as Equation (5) by applying the HSS modeling in the frequency domain.
(5)$$\Delta X_{\mathrm{HSS}}={\left[sI-(\Delta A_{\mathrm{HSS}}-\Delta Q_{\mathrm{HSS}})\right]}^{-1}\Delta B_{\mathrm{HSS}}\Delta U_{\mathrm{HSS}}$$
where Δ***X***_HSS_ and Δ***U***_HSS_ are multi-harmonic vector forms of the state and input variables, respectively (harmonic order is considered to be three to ensure accuracy). Δ***A***_HSS_ and Δ***B***_HSS_ are Toeplitz matrices to perform the frequency domain convolution operation, reflecting the effect of MMC internal and controller multi-harmonic dynamics. Δ***Q***_HSS_ is a diagonal matrix and ***I*** is an identity matrix. The detailed modeling process of the HSS-based MMC is described in [9].

To establish a closed-loop HSS-based MMC model, it is required to consider the control dynamics. When MMC station is connected to the OWF, the AC voltage control strategy is implemented in the WSMMC to guarantee voltage and frequency stability at the point of common coupling (PCC). The WSMMC station adopts the vector control strategy that includes the AC voltage outer controller, phase current inner controller, circulating current suppressing control (CCSC) and frame transformation, as shown in Figure 3.

According to Figure 3, considering the wideband dynamic characteristics of WSMMC, such as multi-frequency coupling and control delay effect, the small-signal insertion indexes generated by the control system can be expressed as follows:(6)$$\left\{\begin{array}{l} \begin{array}{cc} \Delta m_{u} & =G_{\mathrm{de}}(-\Delta m_{f}-\Delta m_{2f})/V_{\mathrm{mdc}}\\ & =-(F_{v}\Delta v_{s}+F_{i}\Delta i_{s})-F_{c}\Delta i_{c} \end{array}\\ \begin{array}{ll} \Delta m_{b} & =G_{\mathrm{de}}(\Delta m_{f}-\Delta m_{2f})/V_{\mathrm{mdc}}\\ & =F_{v}\Delta v_{s}+F_{i}\Delta i_{s}-F_{c}\Delta i_{c} \end{array} \end{array}\right.$$
where
(7)$$\left\{\begin{array}{l} \begin{array}{l} F_{v}(s)=G_{\mathrm{de}}(s)[1-G_{\mathrm{ac}}(s-s_{1})G_{\mathrm{mi}}(s-s_{1})]/V_{\mathrm{mdc}}\\ F_{i}(s)=-G_{\mathrm{de}}(s)[G_{\mathrm{mi}}(s-s_{1})+K_{f}]/V_{\mathrm{mdc}} \end{array}\\ F_{c}(s)=(-G_{\mathrm{ccsc}}(s-s_{1})+K_{2f})/V_{\mathrm{mdc}} \end{array}\right.$$
where Δ***m***_u_ and Δ***m***_b_ are small-signal components of the upper and lower modulation indexes. Δ***m***_f_ and Δ***m***_2f_ are the small-signal modulation voltage at fundamental frequency *f*_1_ and double fundamental frequency 2*f*_1_. *K*_fdq_ = *ω*_1_*L*_eq_ and *K*_2fdq_ = 2*K*_fdq_ are decoupling coefficient matrices at *f*_1_ and 2*f*_1_ in *dq*-frame, respectively, and *L*_eq_ is the equivalent inductance consisting of the MMC arm inductance and the connected transformer inductance. *F*_v_(*s*), *F*_i_(*s*), *F*_c_(*s*) are the transfer functions of AC voltage, phase current and CCSC, in which *G*_ac_(*s*), *G*_mi_(*s*) and *G*_ccsc_(*s*) are the proportional integral (PI) controllers, respectively. *G*_de_ = $e_{}^{-sT_{d}}$ is the total control delay, where *T*_d_ is the sixth-order Pade approximation of the delay time. *V*_mdc_ is the rated DC voltage of the WSMMC.

By substituting Equation (6) into Equation (7) and extracting the voltage and current small-signal components at the perturbation frequency *f*_p_, the transfer function matrix the AC-side of the WSMMC can be derived as
(8)$$\left[\begin{array}{l} \begin{array}{c} 0\\ 0\\ 0 \end{array}\\ \Delta v_{s}-Z_{T3}\Delta i_{s} \end{array}\right]=\left[\begin{array}{cccc} 0 & \cdots & \cdots & 0\\ 0 & \cdots & \cdots & 0\\ 0 & \cdots & \cdots & 0\\ 0 & 0 & 0 & Z_{\mathrm{WSMMC}} \end{array}\right]\left[\begin{array}{l} \Delta i_{c}\\ \Delta v_{\mathrm{cu}}^{\sum }\\ \Delta v_{\mathrm{cb}}^{\sum }\\ \Delta i_{s} \end{array}\right]$$
where *Z*_WSMMC_ is the wideband impedance model of WSMMC. *Z*_T3_ is the equivalent impedance of transformer *T*_3_.

### 2.3. Equivalent Impedance Modeling of OWF

The equivalent impedance model of the OWF is developed in this section by considering the effects of AC submarine cable network and controller dynamics of the wind power converter. The wind power converter control system consists of the phase current controller and a phase-locked loop (PLL), and the small-signal equivalent circuit of the interconnected system is demonstrated in Figure 4.

Based on the harmonic linearization method [10], the wind power converter impedance (*Z*_PM_) can be calculated by considering the frequency coupling effect, which is as follows:(9)$$\left\{\begin{array}{l} Z_{\mathrm{PM}}=\frac{R_{f1}+sL_{f}+K_{\mathrm{pwm}}V_{\mathrm{wdc}}G_{w}}{1-0.5K_{\mathrm{pwm}}V_{\mathrm{wdc}}T_{\mathrm{pll}}\left[I_{1}G_{w}+M_{1}\right]}//(1/sC_{f}+R_{f2})\\ G_{w}(s)=G_{\mathrm{wi}}(s-s_{1})-s_{1}L_{f} \end{array}\right.$$
where *I*_1_ is the amplitude of the phase current. *V*_wdc_ is the DC capacitor voltage of the wind power converter. *M*_1_ is the modulation signal amplitude of the wind power converter. *G*_wi_ is the transfer function of the PI controller of current loop. *s*_1_*L*_f_ is the decoupling term of the current loop. *T*_pll_ is the closed-loop transfer function of the PLL. *K*_pwm_ is the modulation ratio. *L*_f_, *C*_f_, *R*_f1_ and *R*_f2_ are the filter inductance, filter capacitor and parasitic damping resistances, respectively.

The distributed capacitance features of the submarine cable exert significant impact on the MHFRs in the grid-connected system, which are considered using a distributed parameter model here, and the structure is presented in Figure 4, where *k* and *m* represent the head and the end of the cable, respectively. *R*_0_, *l*_0_, *c*_0_ and *g*_0_ denote the resistance, inductance, capacitance and conductance per unit length, respectively.

The frequency domain relationship of voltage and current on both ends of the cable and the high-frequency dynamic model can be expressed as follows:(10)$$\left[\begin{array}{c} {\dot{U}}_{k}\\ {\dot{I}}_{k} \end{array}\right]=\left[\begin{array}{cc} \cosh(\gamma l) & Z_{c}\sinh(\gamma l)\\ \frac{\sinh(\gamma l)}{Z_{c}} & \cosh(\gamma l) \end{array}\right]\left[\begin{array}{c} {\dot{U}}_{m}\\ {\dot{I}}_{m} \end{array}\right]$$
where *γ* = $\sqrt{Z_{0}Y_{0}}$ denotes the cable propagation coefficient. *l* is the cable length. *Z*_c_ = $\sqrt{Z_{0}/Y_{0}}$ represents the cable characteristic impedance, where *Z*_0_ = *r*_0_ + *jwl*_0_ and *Y*_0_ = *jwc*_0_.

The primary concerns in this paper are the voltage and current signals at both ends of the AC cable. Therefore, the medium/high-frequency dynamic model of the AC cable collection network can be illustrated as in Figure 4. The equivalent impedance and admittance of the cable ($Z_{L}^{\prime }$ and $Y_{L}^{\prime }$) can be calculated as follows:(11)$$\left\{\begin{array}{l} Z_{L}^{\prime }=Z_{c}\sinh(\gamma l)\\ Y_{L}^{\prime }=(\cosh(\gamma l)-1)/Z_{c}\sinh(\gamma l)\; \end{array}\right.$$

Furthermore, considering the influence of the distribution characteristics of the AC cable, by converting the parameters on the 0.69 kV and 35 kV sides to those on 220 kV side and combining Equations (9) and (11), the equivalent model of the OWF (*Z*_OWF_) can be obtained as follows:(12)$$\left\{\begin{array}{l} Z_{\mathrm{OWF}}=\frac{Z_{\mathrm{PM}}^{\prime }+(1+Z_{\mathrm{PM}}^{\prime }Y_{L}^{\prime })Z_{L}^{\prime }}{1+Z_{\mathrm{PM}}^{\prime }Y_{L}^{\prime }+Y_{L}^{\prime }[Z_{\mathrm{PM}}^{\prime }+(1+Z_{\mathrm{PM}}^{\prime }Y_{L}^{\prime })Z_{L}^{\prime }]}\\ Z_{\mathrm{PM}}^{\prime }=Z_{\mathrm{PM}}k_{T1}^{2}k_{T2}^{2} \end{array}\right.$$
where *k*_T1_ and *k*_T2_ are the ratio of transformer *T*_1_ and *T*_2_.

### 2.4. Model Validation

To validate the established mathematical model, a detailed simulation model of the MMC-HVDC-connected OWF was built using MATLAB/Simulink and the simulation parameters are given in Table 1. By applying the frequency scanning method, the comparisons between the analytical and simulation results of *Z*_OWF_ and *Z*_WSMMC_ are shown in Figure 5. The analytical model is consistent with the simulation results, confirming the feasibility of the established model for wideband stability analysis.

As shown in Figure 5, the analytical model is consistent with the simulation results, confirming the feasibility of the established impedance model for wideband stability analysis. Figure 5a illustrates that the dynamic performances of the equivalent wind farm impedance model under different submarine cable models are consistent below 800 Hz. When a single-segment π-type cable model was adopted, it failed to reflect the real high-frequency dynamics in the full band. This problem can be solved by increasing the number of series segments of the cable, and the resonant peak number is positively correlated with the series segments. Compared with the π-type model, the distributed parameter model could effectively explain multi-resonance and capacitive negative damping in the medium/high-frequency band caused by the distribution characteristics of the submarine cable, and the features of impedance phase alternating jump become more prominent with the increase in frequency. As observed in Figure 5b, in the phase current controller and harmonic circulating current dynamics, it is easy to introduce resonance peaks in the sub/super-synchronous ranges. The delay effect mainly acts on the medium/high-frequency impedance responses, resulting in a periodic resonance peak and negative damping effects, implying that the impedance interaction points are easily generated in the medium/high-frequency band and thereby affect the stability. In addition, all potential resonance regions can be revealed by adopting the grid-connected model.

## 3. Wideband Small-Signal Stability Mechanism and Constraints Analysis

### 3.1. Active Damping Controller Design

The filters of voltage and current feedforward loop are usually used in the MMC control system to eliminate the negative damping caused by the delay effect. Combined with the feedforward filters and additional damping controller, an active damping controller based on the MHFRs’ suppression method is applied to improve the damping ability of the WSMMC, which is equivalent to paralleling a virtual admittance that is valid for the resonance component. The transfer function and equivalent circuit of the system are given in Figure 6.

As shown in Figure 6, dotted box is the added damping control, red arrows is the affected filtering feedback channel. When considering the damping controller, Equation (7) can be modified as follows:(13)$$\left\{\begin{array}{l} \begin{array}{l} \begin{array}{c} F_{v}^{\prime }(s)=G_{\mathrm{de}}(s)G_{\mathrm{lpf}}^{v}(s)[1-(G_{\mathrm{ac}}(s-s_{1})-\\ \;\;\;\;\;\;\;\;G_{\mathrm{damp}}(s-s_{1}))G_{\mathrm{mi}}(s-s_{1})]/V_{\mathrm{dc}} \end{array}\\ \begin{array}{l} F_{i}^{\prime }(s)=-G_{\mathrm{de}}(s)G_{\mathrm{lpf}}^{i}(s)[G_{\mathrm{mi}}(s-s_{1})+K_{f}]/V_{\mathrm{dc}} \end{array} \end{array} \end{array}\right.$$
where $G_{\mathrm{lpf}}^{v}(s)$ and $G_{\mathrm{lpf}}^{i}(s)$ are the second-order feed-forward LPF functions in voltage and current feedforward channels. *G*_damp_(*s*) is the active damping controller comprising a high-pass filter and a low-pass filter. Their transfer functions can be expressed as follows:(14)$$\left\{\begin{array}{l} G_{\mathrm{lpf}}^{v}(s)=K_{v}({\left(\omega_{{}_{c}}^{v}\right)}^{2}/(s(s+2\xi_{v}\omega_{{}_{c}}^{v})+{\left(\omega_{{}_{c}}^{v}\right)}^{2})\\ G_{\mathrm{lpf}}^{i}(s)=K_{i}({\left(\omega_{{}_{c}}^{i}\right)}^{2}/(s(s+2\xi_{i}\omega_{{}_{c}}^{i})+{\left(\omega_{{}_{c}}^{i}\right)}^{2})\\ G_{\mathrm{damp}}(s)=K_{d}(\omega_{{}_{c}}^{\mathrm{dl}}s)/(s+\omega_{{}_{c}}^{\mathrm{dh}})(s+\omega_{{}_{c}}^{\mathrm{dl}}) \end{array}\right.$$
where *ω*_c_ is the cut-off frequency of LPFs. *ξ* is the damping ratio. *K*_v_ and *K*_i_ are the gain of LPFs, and they are set to 1. *K*_d_ is the virtual gain of *G*_damp_. The superscript “v”, “i”, “dh” and “dl” denote voltage filter, current filter, high-pass filter and low-pass filter, respectively.

### 3.2. Wideband Small-Signal Stability Analysis

The impedance model-based small-signal stability analysis method is an effective tool to investigate the wideband oscillation in power electronic interconnected systems. Based on the refined impedance model of the system, the impedance analysis method is implemented to evaluate the system’s stability by analyzing whether the minor feedback loop gain *Z*_WSMMC_/*Z*_OWF_ satisfies Nyquist criterion [20]; when the resonance intersection of the amplitude–frequency characteristics of *Z*_OWF_ and *Z*_WSMMC_ exists and the phase difference at the intersection is greater than 150°, this indicates that the grid-connected system lacks a stability margin at the resonant frequency. The equivalent small-signal impedance circuit of the interconnected system is shown in Figure 4.

The impedance ratio matrix of the grid-connected system is the equivalent loop gain of the system, $Z_{\mathrm{WSMMC}}(s)/Z_{\mathrm{OWF}}(s),$ and when each subsystem in the wind farm and MMC interconnection system can operate stably independently $T_{m}(s)$, the small interference stability of the system depends on the equivalent loop gain. The actual offshore wind farm or MMC converter station has been guaranteed by the supplier to ensure the rationality of its own parameter design, and when the two are connected to the ideal voltage source or current source separately, they can achieve stable operation; thus, there is no right plane pole in the equivalent loop gain.

By setting one parameter in the impedance expression as a variable, the logarithmic amplitude frequency characteristic of the equivalent loop gain can be obtained; the phase margin PM(s) is then derived as follows:(15)$$PM(s)=180{}^{\circ}+∠T_{m}(j2\pi f_{\mathrm{int}})$$
where $f_{int}$ is the impedance intersection interception frequency. Figure 7 shows the impedance interaction characteristic curves of the grid-connected system under different cable lengths to analyze the influence of the damping control method on the wideband stability. The corresponding parameters are listed in Table 2.

It can be seen from Figure 7 that with an increase in the cable length of 20 km to 60 km and 90 km, the resonance frequency of *Z*_OWF_ shifts forward from 2770 Hz to 1887 Hz and 943 Hz, and the phase margin decreases from 18.07° to −18.85° and −47.22°. According to the impedance stability theory, the negative damping effect intensifies and the potential oscillation frequency turns into the lower-frequency band, which increases the risk of wideband resonances.

After implementing the active damping method on the WSMMC, the high-frequency phase of *Z*_WSMMC_ no longer periodically jumps, remaining in the positive damping interval of ±90° and presenting capacitive characteristics in the high-frequency band. However, the magnitude of *Z*_WSMMC_ near the cut-off frequency changes obviously, indicating that inappropriate controller parameters easily cause negative damping and deteriorate the dynamic performance of the grid-connected system.

On the other hand, the high-frequency stability control method leads to the shift in the resonance peak of *Z*_WSMMC_ in the super-synchronous range, which easily produces resonance intersection points with the wind farm. Despite the high-frequency oscillation suppression strategy being able to eliminate the high-frequency negative damping, it deteriorates the super-synchronous stability and brings new negative damping near the cut-off frequency, especially for a long-distance offshore wind power integration system.

### 3.3. Original Controller Parameter Constraints

The original controllers of the interconnected system play an important role in the wideband dynamic stability, such as fundamental voltage controller or fundamental current controller, and need to be considered in the optimization design process. The dynamic responses of the controller are mainly determined by the proportional coefficient. To investigate the constraints of voltage and current controllers, the corresponding curves of the controllers of *Z*_OWF_ and *Z*_WSMMC_ are plotted in Figure 8, Figure 9 and Figure 10. The black arrow is the change of parameters, and the colored line is the influence of different changes on *Z*_OWF_ and *Z*_WSMMC_.

Figure 8a shows that in the sub/super-synchronous ranges, with an increasing *k*_ip_ of the current controller, the phase jumping point moves forward and the phase margin near 100 Hz decreases, indicating that the amplitude–frequency response tends to be stable. In Figure 8b, the impedance response in the medium/high-frequency ranges is dominated by submarine cable and remains unchanged, demonstrating that the influence of *k*_ip1_ on *Z*_OWF_ concentrates on the sub/super-synchronous-frequency band. To ensure the damping characteristics in the sub/super-synchronous-frequency band, *k*_ip1_ was set in [1.5, 7.5].

It can be found In Figure 8, Figure 9 and Figure 10 that increasing *k*_ip2_ and *k*_vp_ could suppress the resonance peak of the WSMMC in the sub/super-synchronous ranges, and the capacitive negative damping is gradually weakened and the phase margin is increased. In the medium/high-frequency ranges, a significant difference existed in the influence pattern of *k*_ip2_ and *k*_vp_ impedance response of the WSMMC. The amplitude–frequency response increases slightly with the decrease in *k*_ip2_, and a decrease in *k*_ip2_ can decrease the negative damping in the band. Meanwhile, *k*_vp_ could improve the phase margin of *Z*_WSMMC_ and positively correlate with wideband stability. To ensure the damping characteristics in wideband, *k*_vp_ was set in [1.5, 6] and *k*_ip2_ was set in [0.75, 3.75].

### 3.4. Active Damping Controller Parameter Constraints

According to Equation (15), active damping controller is one of the stability dominant factors of the WSMMC, whose parameters (*f*_c_ and *ξ*) need to be carefully designed, where the cut-off frequency and damping ratio of the voltage and current feedforward filters are uniformly chosen and kept unchanged. The corresponding curves of *Z*_WSMMC_ are given in Figure 11 and Figure 12.

As shown in Figure 11, the damping ability in the sub/super-synchronous ranges can be improved by increasing *f*_c_. However, when it becomes relatively high (>1000 Hz), it cannot completely eliminate the negative damping in the high-frequency band caused by the attenuation characteristics of the filters, which is unbeneficial for finding the optimal parameters. In Figure 10, the resonant peak of the amplitude–frequency characteristic curve disappears with a decrease in *ξ* and the negative damping characteristics are effectively suppressed. It should be noted that when *ξ* is too small, it deteriorates the controller stability and dynamic performance of the grid-connected system.

As analyzed above, properly designed controllers of the voltage/current and active damping controllers are essential for a coordinated optimization and the enhancement in the wideband stability, whose proportional coefficients are considered the dominant factors. For the complex model of the grid-connected system, it is necessary to adopt a superior and useful intelligent optimization algorithm when designing the dominant controller parameters for providing positive damping in a wide-frequency band.

## 4. Parameter Optimization Control Strategy Based on IMOSMA

### 4.1. Principle of Improved MOSMA Algorithm

The parameters of the damping controller for wideband resonance suppression are designed such that the stability margin at the impedance intersection point and potential resonance region should be adjusted with positive damping. The SMA is a powerful and advanced population-based optimizer based on the oscillation mode of the slime mold in nature [24]. The population is updated through grabbling, wrapping, and approaching, which can be written as follows:(16)$$\left\{\begin{array}{l} \begin{array}{cc} X(t+1)=rand(u_{b}-l_{b})+l_{b} & \;\;\;\;\;\;\;\;\;\;\;\;\;\;\;\;\;\;\;r<0.03 \end{array}\\ \begin{array}{cc} X(t+1)=X_{b}(t)+v_{b}(WX_{A}(t)-X_{B}(t)) & r<p \end{array}\\ \begin{array}{cc} X(t+1)=v_{c}X(t) & \;\;\;\;\;\;\;\;\;\;\;\;\;\;\;\;\;\;\;\;\;\;\;\;\;\;\;\;\;\;\;\;\;\;\;\;\;\;\;r\geq p \end{array} \end{array}\right.$$
where
(17)$$\left\{\begin{array}{l} W(Sm(i))=\left\{\begin{array}{l} \begin{array}{cc} 1+r\log(\frac{b_{F}-S(i)}{\left(b_{F}-w_{F}\right)}+1) & \mathrm{condition} \end{array}\\ \begin{array}{cc} 1-r\log(\frac{b_{F}-S(i)}{\left(b_{F}-w_{F}\right)}+1) & \mathrm{others} \end{array} \end{array}\right.\\ Sm(i)=sort(S)\\ p=\tan h\left|S(i)-D_{F}\right|\\ \begin{array}{l} v_{b}=\left[-a,a\right],a=\arctan h(-(t/T_{\max})+1)\\ i=1,2,\cdots ,N \end{array} \end{array}\right.$$
where ***X***(*t* + 1) and ***X***(*t*) denote the positions of the slime bacteria at the (*t* + 1)th and *t*th iterations, respectively. ***X***_b_ represents the individual location with the highest food concentration. *rand* and *r* represent the random operators. ***X***_A_(*t*) and ***X***_B_(*t*) are the two randomly selected from the slime individuals at the *t*^th^ iteration. *l*_b_ and *u*_b_ denote the lower and upper search boundaries, respectively. *t* is the current number of iterations. *T*_max_ is the maximum iteration number. *v*_b_ is the vibration parameter. *v*_c_ is a decreasing coefficient from 1 to 0. *rand* and *r* are random values generated between 0 and 1. *S*(*i*) represents the fitness value of the *i*th individual. *D*_F_ represents the optimal fitness value in all iteration results. *N* represents the population size of the slime mold. *W* represents the weight of the slime. “Condition” and “others” represent individuals with fitness values in the first 50% and the last 50%, respectively. *b*_F_ and *w*_F_ represent the optimal fitness value and the worst fitness value in the current iteration result, respectively. *Sm* represents the sequence number of the sorted fitness values. “log” calculation is used to slow down the change rate of the value and stabilize the change in the contraction frequency.

To overcome the drawbacks of the slow convergence rate and insufficient global search ability of traditional MOSMA in solving complex or high-dimensional multi-objective optimization problems, an improved MOSMA is proposed in this paper with the following advantages:(1)A single-dimensional Gaussian mutation strategy is proposed by combining the elite strategy to enhance the global search ability of the SMA, allowing the algorithm to obtain greater convergence and search capabilities. It can be written as follows:
(18)X(t+1)=X(t)+δX(t)   ,   t=1,2,⋯,N
where *δ* is a row vector that obeys the standard normal distribution, whose one random position is a non-zero element.(2)The reference point method [21] is introduced to obtain the next generation of the candidate population for enhancing the diversity of the population. Furthermore, it requires no additional parameters to be set, which is beneficial for solving different problems.(3)The Pareto fronts (PFs) and optimal output solution can be calculated under different practical conditions, implying it is an adaptive algorithm that can be applied to the model of the MMC-HVDC-connected OWF.

Figure 13 depicts a flowchart of the proposed IMOSMA algorithm. The algorithm starts by initializing the controller parameters and randomly generating population *P*_o_. The fitness function of the optimization model is calculated. Then, the positions of the slime mold are updated based on SMA, and *P*_i_ is evaluated using the Gaussian mutation search strategy. After merging *P*_o_ and *P*_i_ to obtain the elite population (*P*_j_), the non-dominated sorting and reference point methods are applied to produce the new population (*P*_i_). This process is repeated until the termination condition is met. Finally, PFs of the optimal solution are obtained using the fuzzy membership-based method [26].

To evaluate the effectiveness of the proposed IMOSMA algorithm, the PFs obtained using the IMOSMA on the well-known MOPs (ZDT4, ZDT6, DTLZ2 and DTLZ7 [24]) are shown in Figure 14. The average values of the performance metrics for IGD and HV values [24] obtained via IMOSMA, NSGA-III and MOSMA are listed in Table 3, where the best performances are shown in bold.

Figure 14 confirms that IMOSMA has stronger convergence and distribution performance. Moreover, it is clear from Table 3 that IMOSMA significantly outperforms the other algorithms for almost all test problems, where the smaller IGD and the greater HV indicate a better solution performance. The statistical results based on the four optimization methods are consistent with the Pareto optimal solutions in Figure 14. Therefore, the comprehensive performance indexes calculated using the proposed IMOSMA method are greater than the prevalent optimization algorithms in terms of the indicators, indicating that it is more suitable for tackling MOPs, such as the controller parameters’ optimization solution of the MMC-HVDC-connected OWF system.

### 4.2. Parameter Optimization Design Based on IMOSMA

On the basis of the obtained influence law of the dominant controller parameters on the system stability, a multi-objective coordination optimization model is established and the parameter optimization scheme is designed using the proposed IMOSMA algorithm. The design scheme can ensure that the key coefficients of those controllers with any initial parameters can improve the wide-frequency impedance characteristics of the grid-connected system after optimization, thereby improving the small-signal stability under different operating conditions.

To facilitate the design of the parameter optimization scheme, the proportional parameters of the dominant controller are used as the optimization variable, such as *k*_ip1_ of wind turbine current controller, *k*_vp_ of MMC voltage controller, *k*_ip2_ of MMC phase current controller and parameters of LPFs. On the other hand, the influence of the high-frequency control strategy on the fault ride-through characteristics of the system needs to be considered, and thus the active damping controller parameters are selected as (620 Hz, 50 Hz, 1.255) to suppress the fault current. The optimization goal is that the amplitude margin and phase margin of the grid-connected system meet the Nyquist stability requirements and the stable region of *k*_ip1_, *k*_vp_, *k*_ip2_; the stability constraints can be obtained as follows:(19)$$\left\{\begin{array}{l} \left|Z_{\mathrm{WSMMC}}(f_{p})\right|<\left|Z_{\mathrm{OWF}}(f_{p})\right|,f_{p}\in \left[10,200\right]\\ ∠Z_{\mathrm{WSMMC}}(f_{p})\in \left[-90{}^{\circ},90{}^{\circ}\right],f_{p}\in \left[60,4000\right]\\ k_{\mathrm{ip}1}\in \left[1.5,7.5\right]\\ k_{\mathrm{vp}}\in \left[1.5,7.5\right]\\ k_{\mathrm{ip}2}\in \left[0.75,3.75\right] \end{array}\right.$$
where *f*_p_ denotes the injected perturbation frequency. |*Z*_WSMMC_(*f*_p_)| and ∠*Z*_WSMMC_(*f*_p_) are magnitude and phase of *Z*_WSMMC_ at *f*_p_, respectively. |*Z*_OWF_(*f*_p_)| is the phase of *Z*_OWF_ at *f*_p_.

To filter medium/high-frequency resonance components and mitigate the instability risks originated from the interaction between the attenuation characteristics of the LPFs and the dominant controller dynamics, the objective functions *F*_1_, *F*_2_ are developed as follows:(20)$$\left\{\begin{array}{cc} \max & F_{1}=\min\left\{\left|\left|G_{\mathrm{lpf}}^{v}(f_{p})\right|,\left|G_{\mathrm{lpf}}^{i}(f_{p})\right|\right|f_{p}=80\mathrm{Hz}\right\}\\ \min & F_{2}=\max\left\{\left|\left|G_{\mathrm{lpf}}^{v}(f_{p})\right|,\left|G_{\mathrm{lpf}}^{i}(f_{p})\right|\right|f_{p}=2000\mathrm{Hz}\right\} \end{array}\right.$$
where *F*_1_ and *F*_2_ represent the minimum and maximum magnitude absolute values of the LPFs at 80 Hz and 2000 Hz, respectively.

To compare the performance of the proposed multi-objective optimization model, taking the MMC-HVDC grid-connected system of the proposed four-unit equivalent offshore wind farm as the object, the proposed IMOSMA algorithm is used to solve the optimal controller parameter combination, and the number of iterations is set as Tmax = 500 and the population size as N = 200. Figure 15 shows the distribution of the PFs under different algorithms where the optimal solution is marked with a red dashed line. The corresponding optimal controller parameters are listed in Table 4. Figure 16 shows the optimization results of impedance responses of the LPFs [23,24].

Figure 15 shows that the solutions obtained using the proposed method converge to the PF with better quality and evenness than the others. Meanwhile, the feasible region of the controller parameters for stable operation could be derived corresponding to the PF. However, *F*_2_ increases as *F*_1_ increases, indicating that the objective values cannot simultaneously achieve the optimal result. Combined with the fuzzy membership-based method, the optimal compromise solution is obtained and presented in Figure 15.

As shown in Figure 16, after parameter optimization based on the proposed method, the optimal values of *F*_2_ and *F*_1_ are 0.9307 and 0.0985 from $G_{\mathrm{lpf}}^{i}$, respectively, where the decrease in the LPFs’ magnitude is of less than 10% below 80 Hz but the decrease is of over 90% above 2000 Hz. The optimization solutions are matched with the analysis results shown in Figure 15. 

Figure 17 gives a comparison of the wideband small-signal stability with and without parameter optimization. After adopting the IMOSMA-based controller design method, the impedance resonance peaks and potential risky region in the sub/super-synchronous ranges are eliminated, and all the phase margins are greater than 30°, meeting the stability requirements.

It can be seen from Figure 18 that when the length of the submarine cable Lh is increased to 60 km, the frequency bands of the wind farm’s multiple resonance peaks and periodic capacitive phase characteristics are shifted forward, the high-frequency negative damping caused by the control delay in the MMC is carried out before the control strategy is put into operation, and the optimization of the wind farm side parameters can suppress the capacitive fluctuations of the wind farm resonance peak and the medium- and high-frequency bands.

When the feedforward improvement link is used on the MMC side, there are still a large number of resonant peaks and medium- and high-capacitive fluctuations in the impedance characteristics, and the improvement effect of impedance characteristics and dynamic performance is more significant after adding parameter optimization. The detailed design process of the proposed control strategy is illustrated in Figure 19. Based on the wideband impedance modeling of the system, the stability constraints can be derived by considering the wideband dynamics of the controllers. Then, the feasible stability region of the controller parameters can be obtained by applying the IMOSMA algorithm. Finally, a comparative analysis of the simulation and experiments is conducted for verification.

## 5. Simulation and Experimental Verification

### 5.1. Simulation Results

Based on the detailed simulation model of grid-connected system with the 90 km submarine cable, the transmission network was used as the simulation example to verify the effectiveness of the proposed optimization approach. The simulation results of the AC voltage and current in the per-unit system (*i*_pcc_pu_ and *v*_pcc_pu_) with and without launching the coordinated optimization control strategy are shown in Figure 20.

It can be seen from Figure 20 that before adding the parameter optimization control strategy, when the disturbance is added, due to the lack of phase stability margin, there are harmonic components in the current and voltage curves, and the waveform is more violently jitter. *I*_pcc_pu_ and *v*_pcc_pu_ contain harmonic components within 0.81 s, where the harmonic component of *i*_pcc_pu_ occurs at 948 Hz. After optimizing the controller parameters at 0.81s, the oscillation of the current voltage decreases and the resonances are suppressed within one cycle. The system restores stable operation, which verifies the theory analysis results in Figure 7. Hence, the influences of the distribution characteristic of submarine cable and the delay effect on the small-signal stability of the system are eliminated via the proposed coordinated optimization method, and the dynamics in the sub/super-synchronous ranges are performed well, realizing the wideband resonance suppression of the system. It is proved that the proposed strategy can suppress the phenomenon of medium- and high-frequency dynamic oscillation, and provide technical support for promoting flexible long-distance transmission and large-scale development of offshore wind farms and ensuring the safety and stability of grid-connected systems.

### 5.2. Experimental Results

A control-hardware-in-the-loop (CHIL) experimental platform was built to further validate the proposed strategy, as shown in Figure 21. Figure 22 shows the experimental and fast Fourier transformation (FFT) results of the phase current *i*_pcc_a_ and phase voltage *v*_pcc_a_ at the PCC point of the system under 60 km cable (case I and case II) and 90 km cable (case III and case IV) before and after adopting the proposed strategy.

It is shown in Figure 22a,c that before adopting the proposed optimization method, in *i*_pcc_a_ and *v*_pcc_ab_, there exists a significant high-frequency resonance component. According to the FFT result of *i*_pcc_a_ in Figure 23a,b, when the length of the submarine cable is 60 km and 90 km, the dominant resonance frequency occurs at 1890 Hz and 945 Hz, whose THDs are 21.64% and 19.33%, respectively. The results are also compatible with the impedance stability analysis results in Figure 7. It can be seen that the MMC-HVDC-connected OWF easily leads to harmonic resonance due to the interactions between the capacitance effect of submarine cable, the delay effect of MMC and the negative damping characteristics of the control system.

As shown in Figure 22b,d, after adopting the damping controller with optimized controller parameters applied in both the wind farm and the MMC system, the grid-connected system can achieve wideband stable operation under different cable length configurations, and the THDs are 0.79% and 0.96%, respectively, which is consistent with the results in Figure 17. This indicates that the control system with the optimized parameters that obey stability constraints have a strong damping ability that can suppress the harmonic resonance, and enhance the wideband stability and the power quality of the system, especially for the long-distance wind power integration system. The experimental results further verify the effectiveness of the proposed optimization control strategy.

## 6. Conclusions

Based on the impedance stability analysis theory, this paper investigates the wideband stability mechanism of an MMC-HVDC-connected OWF and an IMOSMA-based coordinated optimization control strategy is proposed. The following conclusions can be obtained:(1)The established distribution parameter model is applicable to the long submarine cable network and reflects the periodic negative damping dynamics in the medium/high response, and the control delay effect has a similar influence in MMC. This solves the problem of the analysis of medium- and high-frequency oscillation characteristics being inaccurate due to ignoring the dynamic characteristics of medium and high frequencies in the existing methods.(2)The proportional coefficient of fundamental frequency voltage and current controllers in MMC and the phase current controller in the wind power converter, and the active damping controllers are the dominant factors of the wideband stability. The parallel virtual admittance control strategy could eliminate the negative damping behaviors in the medium/high-frequency band while deteriorating the sub/super-synchronous damping performance and intensifying the risk of super-synchronous resonances. The elucidated mechanism of medium- and high-frequency dynamic stability provides theoretical support for research on the strategy of improving the dynamic stability of medium and high frequency.(3)The proposed IMOSMA optimization algorithm outperforms the others, improving the population quality and global search ability of the algorithm, and the coordinated control strategy could enhance the adaptability of controller parameters and the wideband damping ability of the grid-connected system for meeting the wideband stability demand. The applicable regions of the controller parameters were also derived. The simulation and experimental results verify the correctness of the stability analysis conclusion and feasibility of the proposed control strategy.

## Figures and Tables

**Figure 1 sensors-24-00139-f001:**
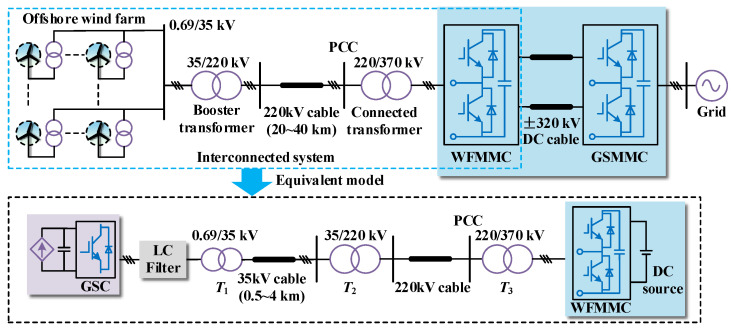
Structure diagram of MMC-HVDC-connected OWF and interconnected system.

**Figure 2 sensors-24-00139-f002:**
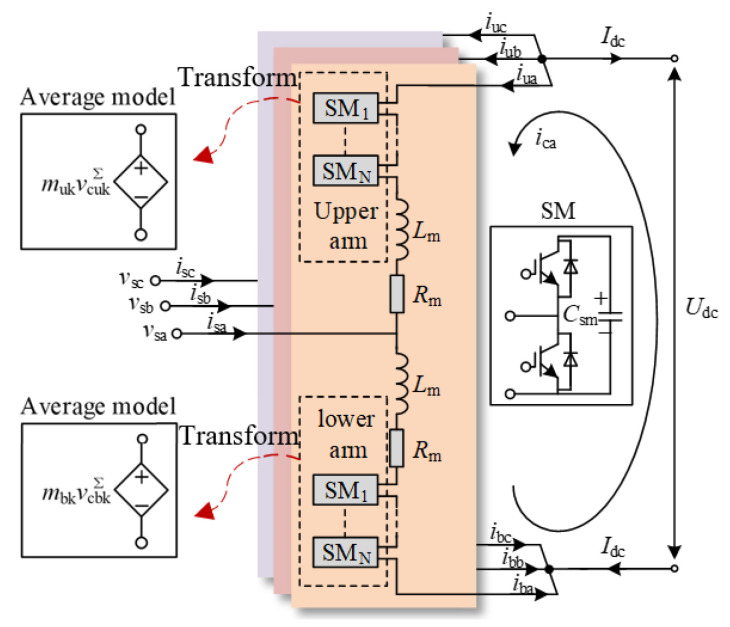
Three-phase main circuit configuration of MMC.

**Figure 3 sensors-24-00139-f003:**
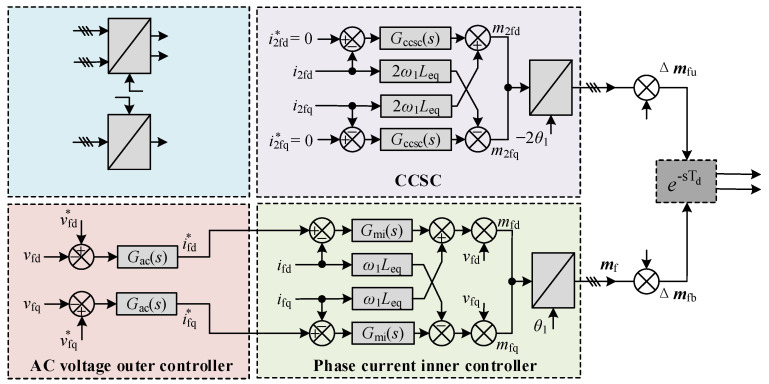
Schematic of the control system of WSMMC.

**Figure 4 sensors-24-00139-f004:**
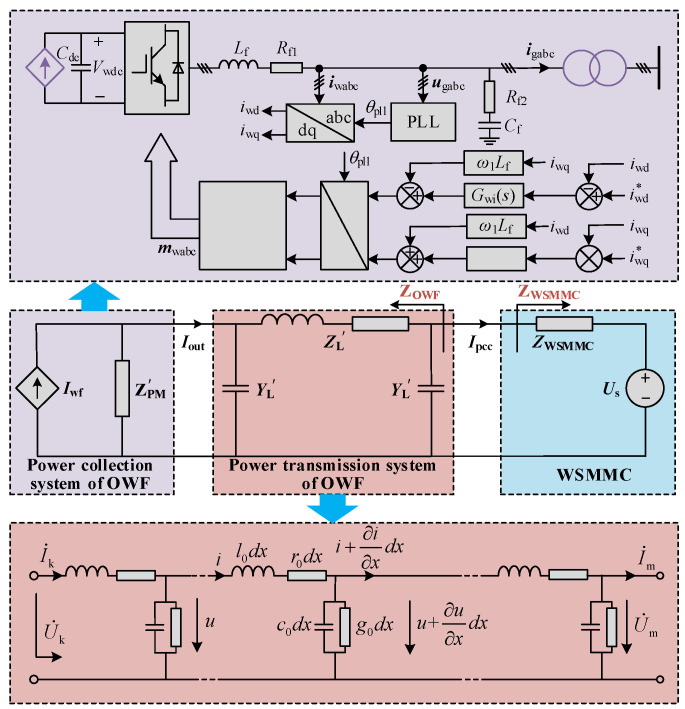
Equivalent small-signal circuit of OWF and interconnected system.

**Figure 5 sensors-24-00139-f005:**
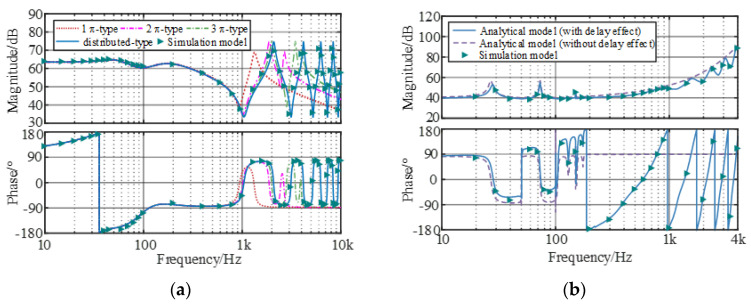
Frequency response comparisons of theoretical impedance model and simulated results. (**a**) *Z*_OWF_ and (**b**) *Z*_WSMMC_.

**Figure 6 sensors-24-00139-f006:**
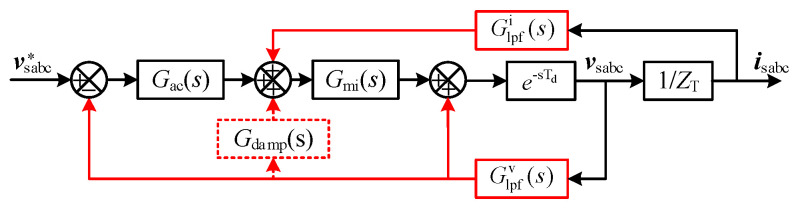
Schematic of active damping controller of WSMMC.

**Figure 7 sensors-24-00139-f007:**
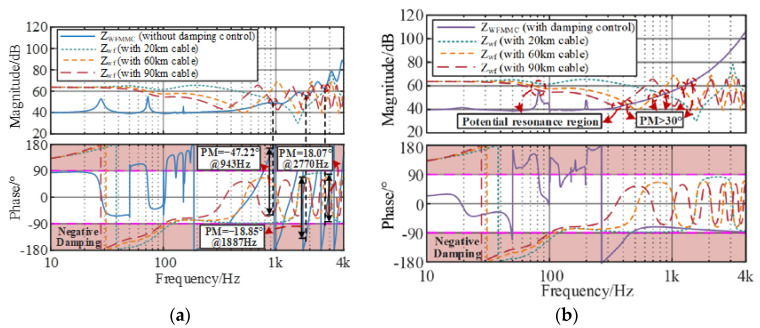
Wideband stability analysis under different cable lengths. (**a**) Without damping control. (**b**) With damping control.

**Figure 8 sensors-24-00139-f008:**
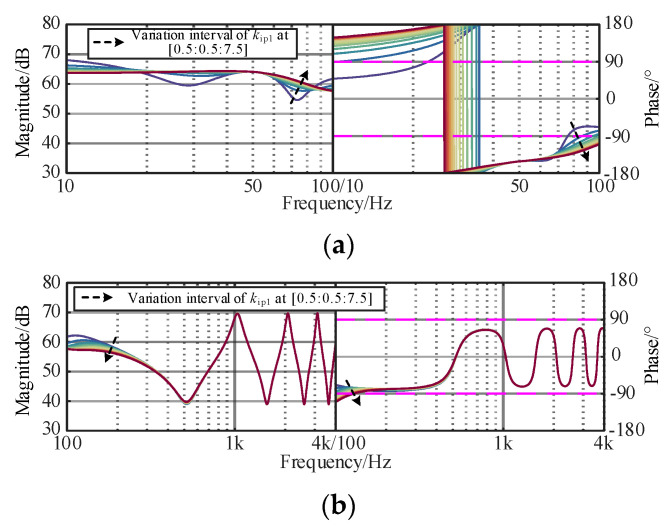
Influence of *k*_ip1_ of wind turbine current controller on *Z*_OWF_. (**a**) Sub/super-synchronous-frequency band. (**b**) Medium/high-frequency band.

**Figure 9 sensors-24-00139-f009:**
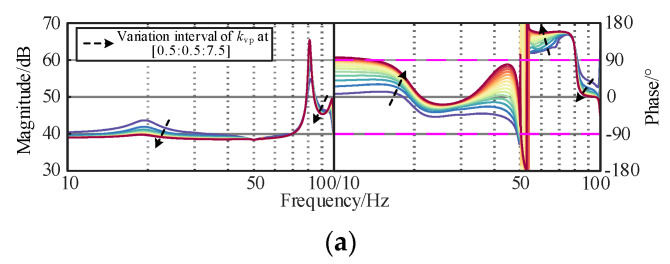
Influence of *k*_vp_ of *Z*_WSMMC_. (**a**) Sub/super-synchronous-frequency band. (**b**) Medium/high-frequency band.

**Figure 10 sensors-24-00139-f010:**
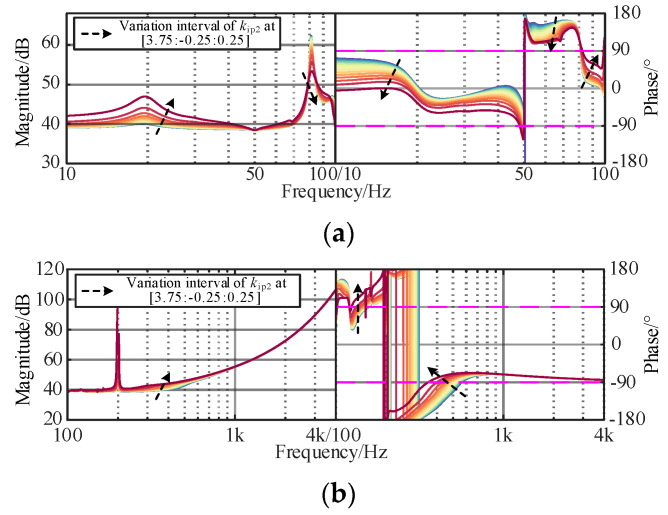
Influence of *k*_ip2_ of *Z*_WSMMC_. (**a**) Sub/super-synchronous-frequency band. (**b**) Medium/high-frequency band.

**Figure 11 sensors-24-00139-f011:**
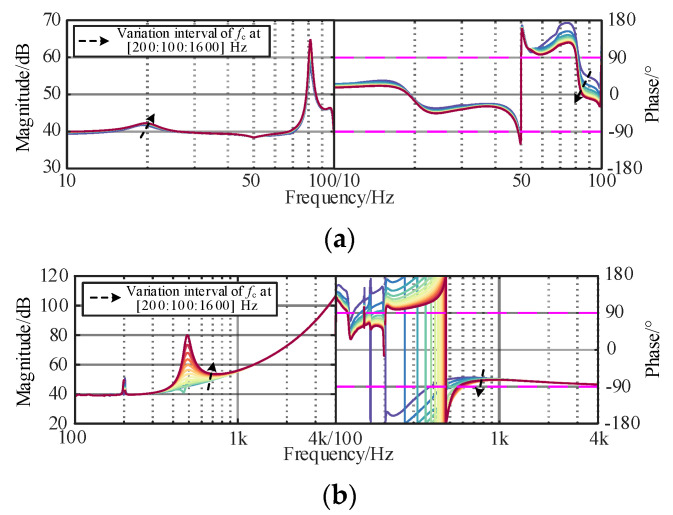
Influence of *f*_c_ of MMC damping controller on *Z*_WSMMC_. (**a**) Sub/super-synchronous-frequency band. (**b**) Medium/high-frequency band.

**Figure 12 sensors-24-00139-f012:**
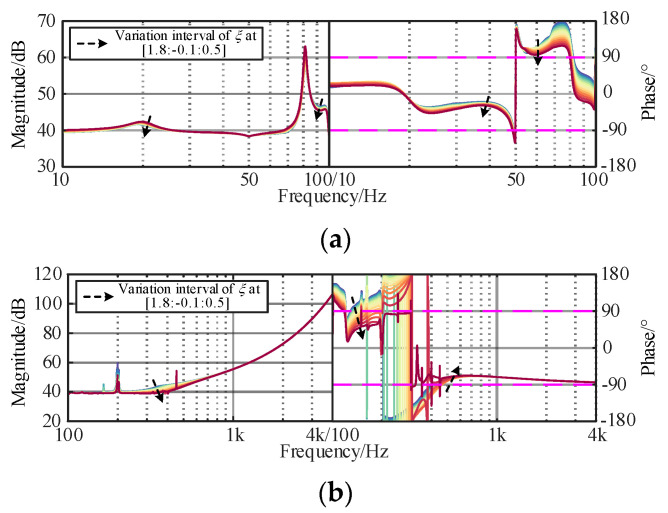
Influence of *ξ* of MMC damping controller on *Z*_WSMMC_. (**a**) Sub/super-synchronous-frequency band. (**b**) Medium/high-frequency band.

**Figure 13 sensors-24-00139-f013:**
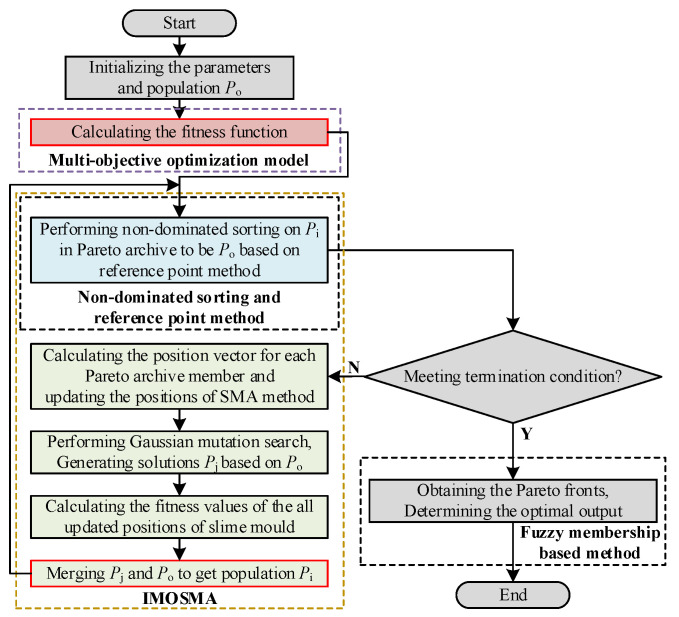
Flowchart of the proposed IMOSMA algorithm.

**Figure 14 sensors-24-00139-f014:**
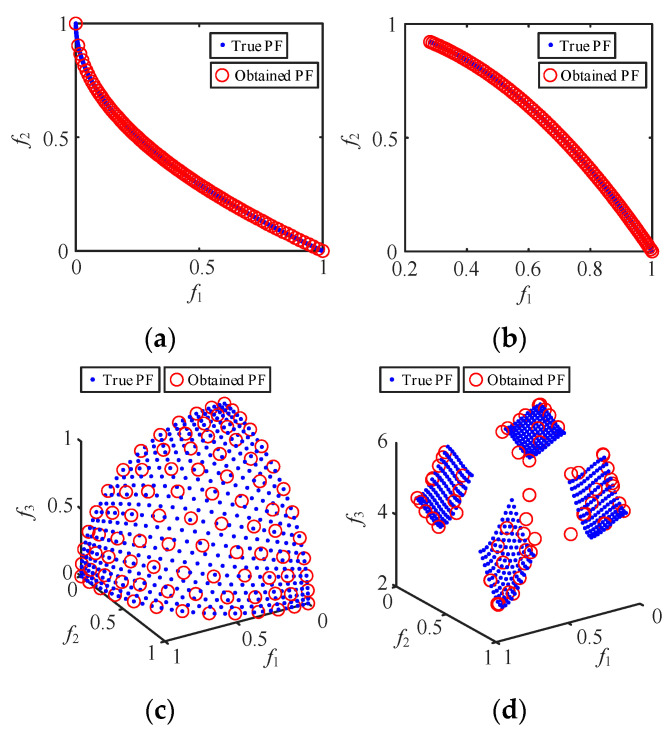
Results of PFs obtained via IMOSMA for the analyzed MOPs. (**a**) ZDT4. (**b**) ZDT6. (**c**) DTLZ2. (**d**) DTLZ7.

**Figure 15 sensors-24-00139-f015:**
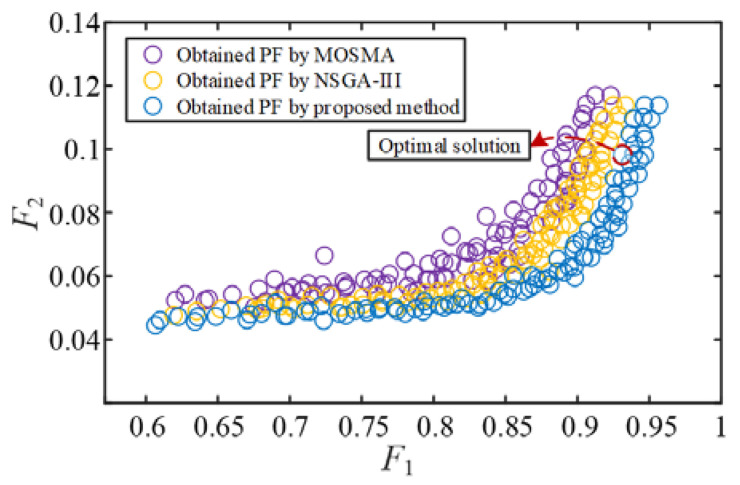
PFs under different optimization algorithms.

**Figure 16 sensors-24-00139-f016:**
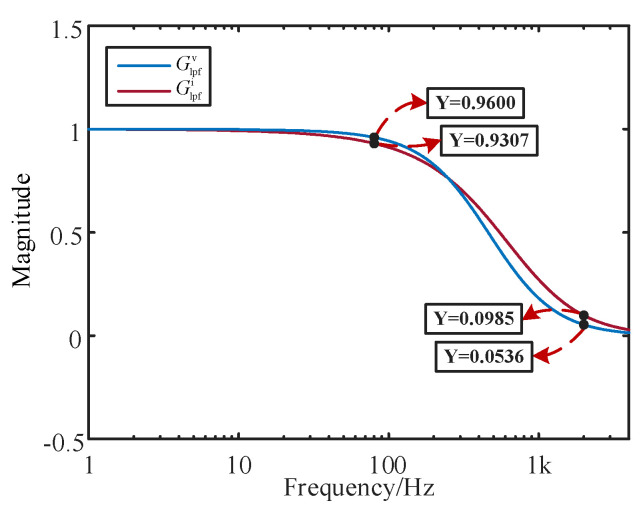
Impedance responses of the LPFs.

**Figure 17 sensors-24-00139-f017:**
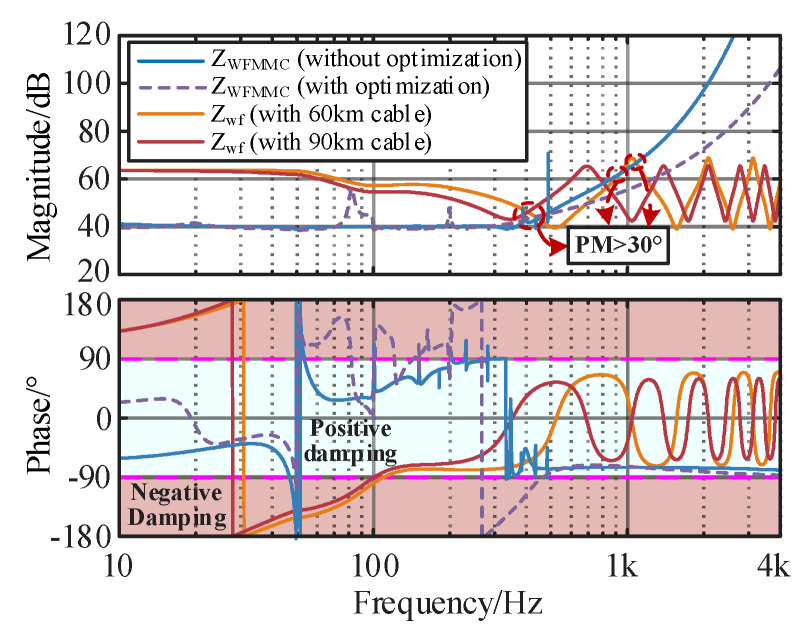
Comparison of wideband impedance characteristics.

**Figure 18 sensors-24-00139-f018:**
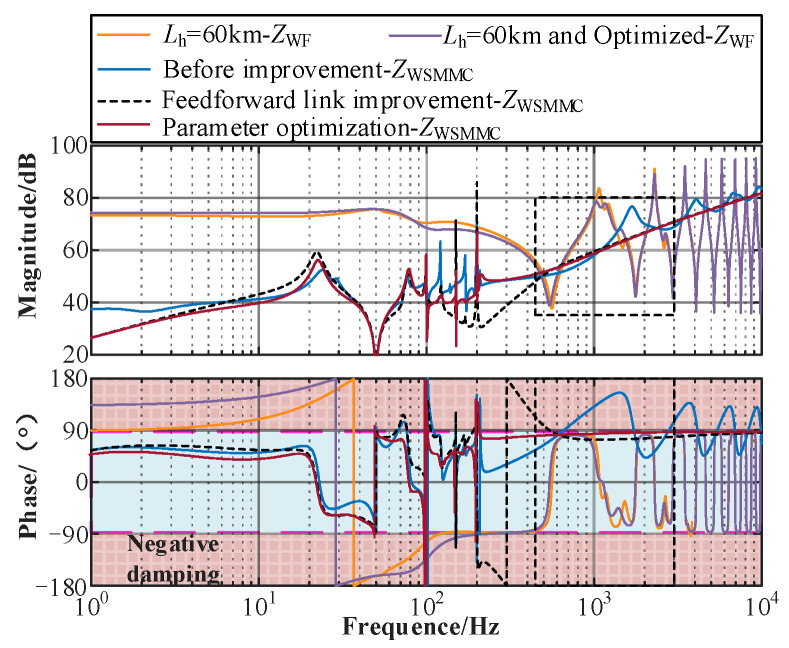
Comparison of wideband impedance characteristics under different strategies.

**Figure 19 sensors-24-00139-f019:**
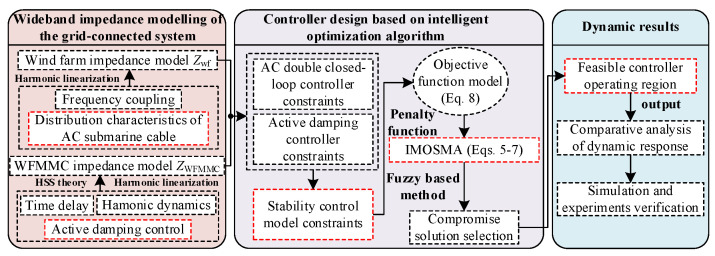
Design process of the proposed control strategy.

**Figure 20 sensors-24-00139-f020:**
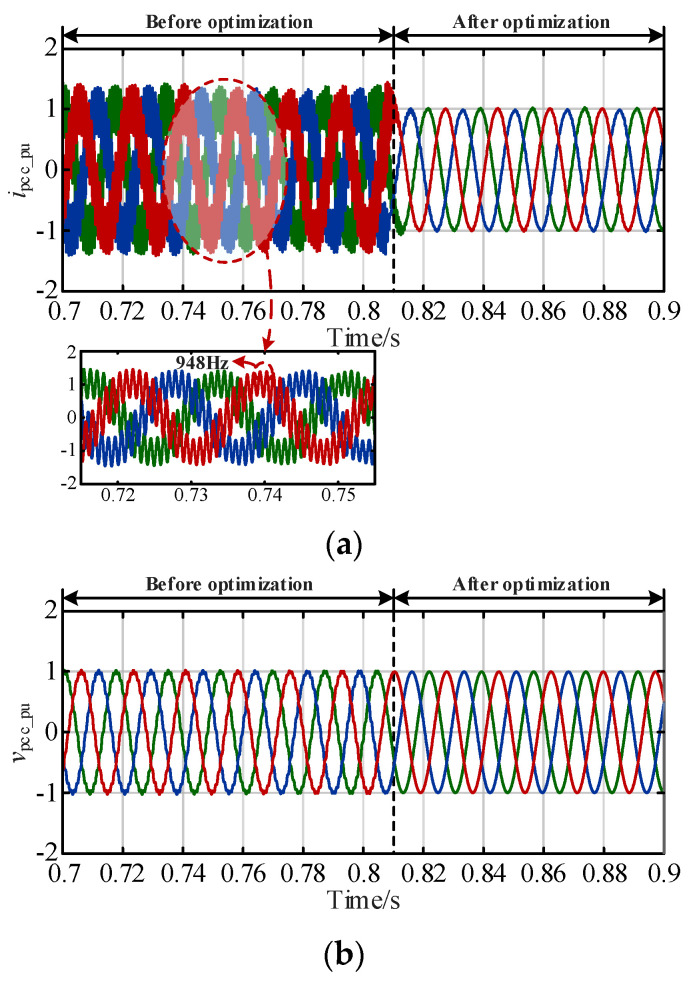
Comparison of simulation results. (**a**) *i*_pcc_abc_. (**b**) *v*_pcc_abc._

**Figure 21 sensors-24-00139-f021:**
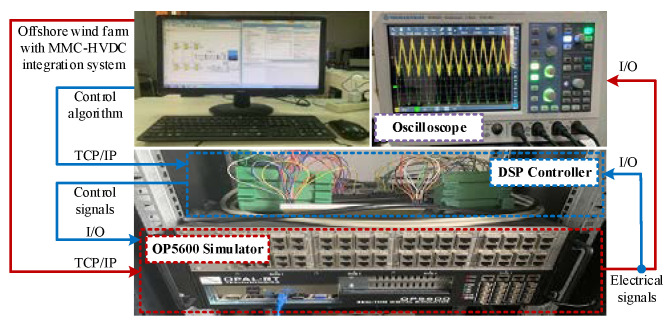
Prototype of CHIL experiment platform.

**Figure 22 sensors-24-00139-f022:**
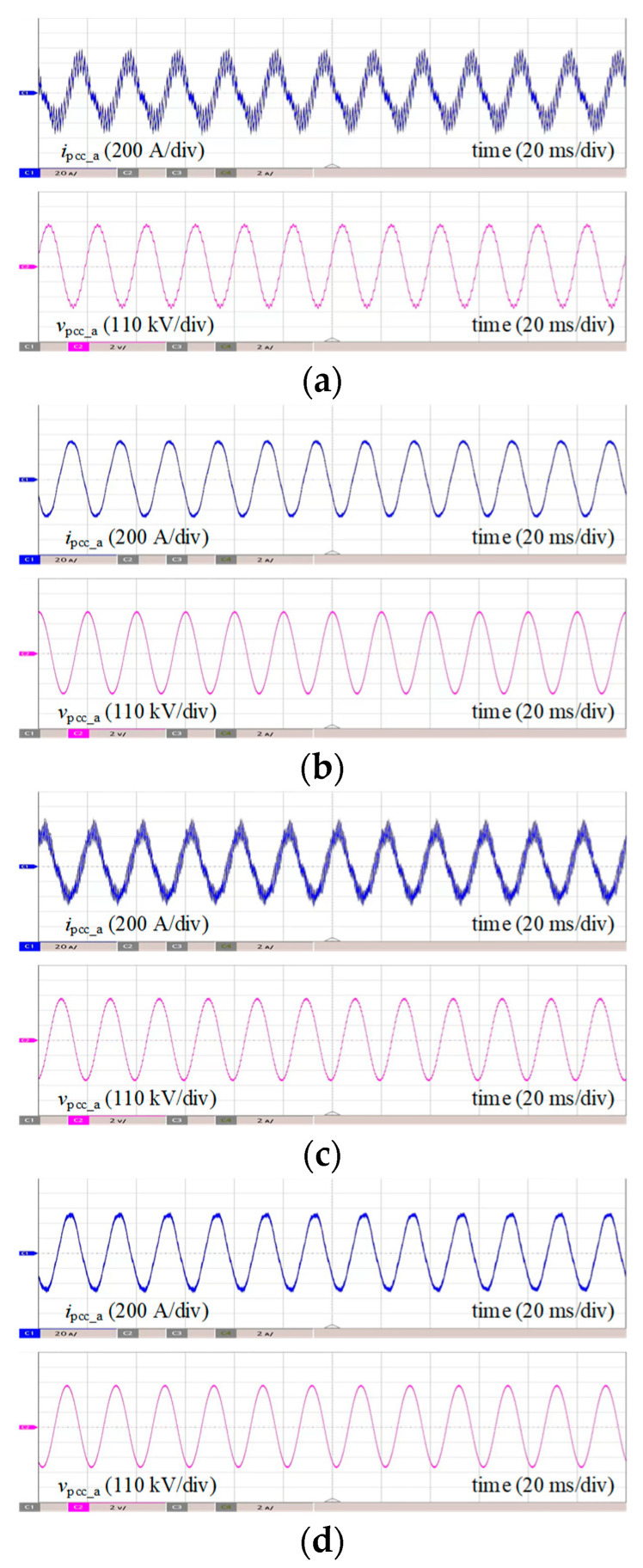
Comparison of experimental results (*i*_pcc_a_ and *v*_pcc_ab_). (**a**) Case I. (**b**) Case II. (**c**) Case III. (**d**) Case IV.

**Figure 23 sensors-24-00139-f023:**
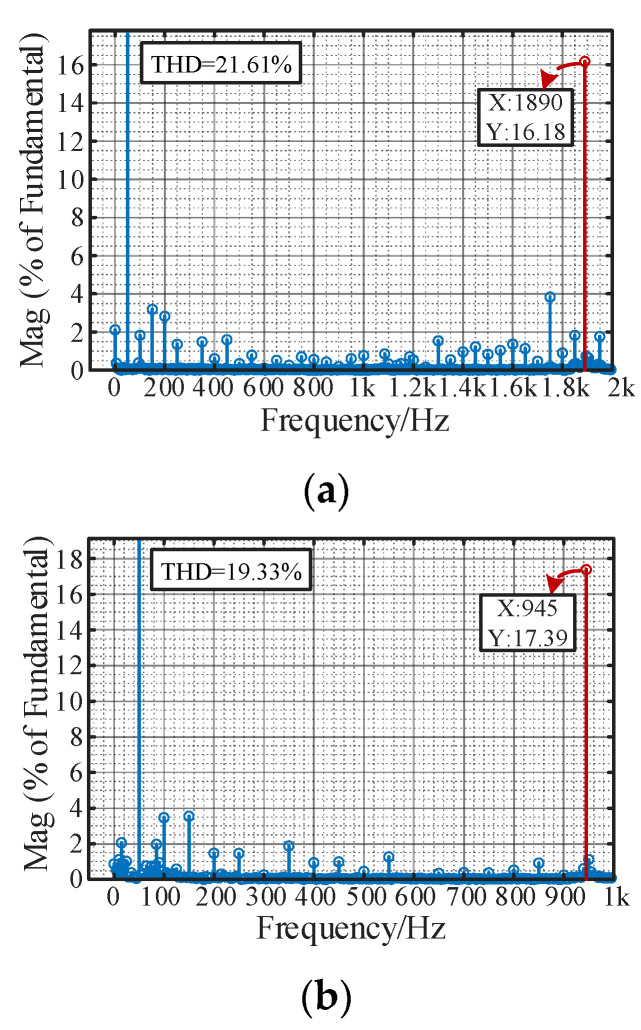
Comparison of FFT results (*i*_pcc_a_). (**a**) Case I. (**b**) Case III.

**Table 1 sensors-24-00139-t001:** Parameters of MMC-HVDC-connected OWF.

Model	Parameter	Value
OWF	Nominal power	*S*_base1_ = 400 MW
	Nominal AC voltage	*V*_n_ = 0.69 kV
	Nominal DC voltage	*V*_wdc_ = 1.2 kV
	Phase current regulator	*k*_ip1_ = 3.107, *k*_ii1_ = 120
	PLL regulator	*k*_pp_ = 12.003, *k*_pi_ = 180
AC cable collection network	35 kV cable resistance	*R*_35_ = 0.365 Ω/km
	35 kV cable inductance	*L*_35_ = 0.503 mH/km
	35 kV cable capacitance	*C*_35_ = 0.117 μF/km
	220 kV cable resistance	*R*_220_ = 0.064 Ω/km
	220 kV cable inductance	*L*_220_ = 0.427 mH/km
	220 kV cable capacitance	*C*_220_ = 0.139 μF/km
WSMMC	AC system line voltage	*V*_s_ = 370 kV
	DC Nominal voltage	*V*_mdc_ = 640 kV
	Number of submodules	*N*_s_ = 200
	Submodule capacitance	*C*_sm_ = 8 mF
	Arm inductance	*L*_arm_ = 80 mH
	Arm resistance	*R*_arm_ = 1 Ω
	Delay time	*T*_d_ = 550 μs
	AC voltage regulator	*k*_vp_ = 1.102, *k*_vi_ = 30
	Phase current regulator	*k*_ip2_ = 2.816, *k*_ip2_ = 50
	CCSC	*k*_cp_ = 38.805, *k*_ci_ = 1500

**Table 2 sensors-24-00139-t002:** Parameters of filters and active damping controller.

Controllers	Parameters	Value
$G_{\mathrm{lpf}}^{v}(s)$	$\left(f_{{}_{c}}^{i},\xi_{i},K_{i}\right)$	(900 Hz, 0.707, 1)
$G_{\mathrm{lpf}}^{i}(s)$	$\left(f_{{}_{c}}^{v},\xi_{v},K_{v}\right)$	(900 Hz, 0.707, 1)
*G*_damp_(*s*)	$\left(f_{{}_{c}}^{\mathrm{dh}},f_{{}_{c}}^{\mathrm{dl}},K_{d}\right)$	(30 Hz, 400 Hz, 1)

**Table 3 sensors-24-00139-t003:** Results of the multi-objective algorithm.

MOPs	IMOSMA		NSGA-III		MOSMA	
	IGD (10^−3^)	HV (10^−1^)	IGD (10^−3^)	HV (10^−1^)	IGD (10^−3^)	HV (10^−1^)
ZDT4	**3.966**	**7.199**	17.83	7.053	358.1	4.719
ZDT6	**3.023**	**3.889**	3.689	3.873	29.36	3.600
DTLZ2	**52.92**	**5.591**	152.0	5.148	109.8	4.910
DTLZ7	84.75	**2.686**	**79.48**	2.658	151.1	2.300

**Table 4 sensors-24-00139-t004:** Optimal solution comparison of analyzed algorithms.

Performance	Optimized Controller Parameters				
	*k* _ip1_	*k* _vp_	*k* _ip2_	$\left(f_{{}_{c}}^{v},\xi_{v}\right)$	$\left(f_{{}_{c}}^{i},\xi_{i}\right)$
Initial value	3.107	1.102	2.812	(900 Hz, 0.707)	(900 Hz, 0.707)
MOSMA in [24]	6.306	4.780	0.791	(480 Hz, 0.712)	(1378 Hz, 2.251)
NSGA-III in [23]	4.889	5.331	0.384	(338 Hz, 0.574)	(1038 Hz, 1.811)
Proposed method	5.124	5.818	0.515	(496 Hz, 0.531)	(1285 Hz, 1.529)

## Data Availability

Data are contained within the manuscript.

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
