# Peer review of "Small-Signal Stability Analysis and MOSMA-Based Optimization Control Strategy of OWF with MMC-HVDC Grid Connection"

_sensors, 2023, doi:10.3390/s24010139_

Round 1
Reviewer 1 Report
Comments and Suggestions for Authors
As the reviewer of this paper, some major revisions comments I would suggest:
1. The paper could benefit from clearer delineation of the methodologies employed in the impedance analysis and the subsequent stability investigations. Define the specific steps and processes of impedance modeling, stability analysis, and control strategy development with more explicit clarity. This will help readers follow the technical intricacies more comprehensively.
2. While the paper mentions its contributions, it would be beneficial to explicitly outline these contributions at the beginning or in a dedicated section. Highlighting these contributions will assist readers in understanding the novel aspects and significant advancements presented in the study.
3. Incorporate a more detailed comparative analysis section. Compare the proposed methodology with existing approaches or methodologies in the field. Highlight the strengths and weaknesses of the proposed approach concerning previously established techniques, elucidating the advancements or improvements achieved in the study.
4. While the paper mentions the verification through simulation and experimental results, it lacks a thorough discussion and interpretation of these findings. Provide a more comprehensive discussion of the simulation and experimental outcomes, emphasizing how they validate or support the proposed methodology and its effectiveness in enhancing small-signal stability.
5. Expand upon the discussion regarding the practical implications of the proposed control strategy and impedance analysis for real-world applications. Discuss how these findings can be translated into practical implementation in offshore wind farm systems utilizing MMC-HVDC connections, addressing practical challenges and feasibility.
6. The topics related to the stability of the power systems, such as Damping of subsynchronous resonance in utility DFIG-based wind farms using wide-area fuzzy control approach; Observer-based predictive control of nonlinear clutchless automated manual transmission for pure electric vehicles: An LPV approach, should be considered in the literature.
7. Lastly, consider refining the language and structure for better readability. Ensure that technical terms are adequately explained, and the flow of information is smooth and coherent. This will aid in making the complex technical content more accessible to a wider audience.
Comments on the Quality of English LanguageIt is suitable according to the quality of English.
Reviewer 2 Report
Comments and Suggestions for Authors

Carefully proofreading paper to correct minor grammatical issues and insure a polished presentation.
